# Impaired Ca^2+^ Sensitivity of a Novel GCAP1 Variant Causes Cone Dystrophy and Leads to Abnormal Synaptic Transmission Between Photoreceptors and Bipolar Cells

**DOI:** 10.3390/ijms22084030

**Published:** 2021-04-14

**Authors:** Valerio Marino, Giuditta Dal Cortivo, Paolo Enrico Maltese, Giorgio Placidi, Elisa De Siena, Benedetto Falsini, Matteo Bertelli, Daniele Dell’Orco

**Affiliations:** 1Department of Neurosciences, Biomedicine and Movement Sciences, Section of Biological Chemistry, University of Verona, 37129 Verona, Italy; valerio.marino@univr.it (V.M.); giuditta.dalcortivo@univr.it (G.D.C.); 2MAGI’S Lab S.R.L., 38068 Rovereto, Italy; paolo.maltese@assomagi.org (P.E.M.); matteo.bertelli@assomagi.org (M.B.); 3Fondazione Policlinico Universitario “A. Gemelli”, IRCCS, 00168 Rome, Italy; giorgioplacidi@libero.it (G.P.); elisa.desiena@yahoo.com (E.D.S.); 4Università Cattolica del Sacro Cuore, 00168 Rome, Italy; 5MAGI Euregio, 39100 Bolzano, Italy

**Keywords:** GUCA1A, neuronal calcium sensor, phototransduction, synaptic transmission, cone dystrophy, guanylate cyclase, calcium binding proteins, bipolar cells, photoreceptors, retinal degeneration

## Abstract

Guanylate cyclase-activating protein 1 (GCAP1) is involved in the shutdown of the phototransduction cascade by regulating the enzymatic activity of retinal guanylate cyclase via a Ca^2+^/cGMP negative feedback. While the phototransduction-associated role of GCAP1 in the photoreceptor outer segment is widely established, its implication in synaptic transmission to downstream neurons remains to be clarified. Here, we present clinical and biochemical data on a novel isolate GCAP1 variant leading to a double amino acid substitution (p.N104K and p.G105R) and associated with cone dystrophy (COD) with an unusual phenotype. Severe alterations of the electroretinogram were observed under both scotopic and photopic conditions, with a negative pattern and abnormally attenuated b-wave component. The biochemical and biophysical analysis of the heterologously expressed N104K-G105R variant corroborated by molecular dynamics simulations highlighted a severely compromised Ca^2+^-sensitivity, accompanied by minor structural and stability alterations. Such differences reflected on the dysregulation of both guanylate cyclase isoforms (RetGC1 and RetGC2), resulting in the constitutive activation of both enzymes at physiological levels of Ca^2+^. As observed with other GCAP1-associated COD, perturbation of the homeostasis of Ca^2+^ and cGMP may lead to the toxic accumulation of second messengers, ultimately triggering cell death. However, the abnormal electroretinogram recorded in this patient also suggested that the dysregulation of the GCAP1–cyclase complex further propagates to the synaptic terminal, thereby altering the ON-pathway related to the b-wave generation. In conclusion, the pathological phenotype may rise from a combination of second messengers’ accumulation and dysfunctional synaptic communication with bipolar cells, whose molecular mechanisms remain to be clarified.

## 1. Introduction

The signaling machinery underlying the phototransduction cascade is finely regulated by Ca^2+^ and cyclic guanosine monophosphate (cGMP), which constitute strictly interconnected second messengers [1,2]. The light-induced activation of phosphodiesterase 6 catalyzes the hydrolysis of cGMP, which causes cGMP-gated channels (CNG) to transiently close and lead to a hyperpolarization of the cell membrane which is sensed at the photoreceptor synaptic terminal. As a consequence of such closure and the light-independent extrusion of Ca^2+^ from the Na^+^/Ca^2+^, K^+^-exchanger, the concentration of Ca^2+^ in the photoreceptor outer segments drops down to below 100 nM in the light [3]. The neuronal calcium sensor guanylate cyclase-activating protein 1 (GCAP1) is responsible for detecting subtle changes in Ca^2+^ concentration in both rods and cones, thus constituting a fine modulator of retinal guanylate cyclase (GC) activity. Specifically, the Ca^2+^-loaded GCAP1 conformation in the dark prevents the activation of GC. Following the light-induced drop in [Ca^2+^], GCAP1 replaces Ca^2+^ ions for Mg^2+^ [4,5]. Mg^2+^-GCAP1 thus acquires a conformation that stimulates the synthesis of cGMP by GC to rapidly restore dark-adapted cell conditions by reopening the CNG channels [6,7]. Two isoforms of retinal GC have been found in photoreceptors, namely, GC1 (or RetGC-1, GC-E) and GC2 (RetGC-2, GC-F), although the latter produces less than 30% of cGMP in murine retina [8] and is 25-fold less abundant than GC1 in bovine rod outer segments [9].

To date, more than twenty point mutations identified in *GUCA1A*, the gene coding for GCAP1, have been associated with various forms of autosomal dominant cone dystrophies (adCOD) [10,11,12,13,14,15,16,17,18,19,20,21,22,23,24], a class of severe retinal degeneration diseases characterized by central vision loss, impaired color vision, and photophobia [25]. A thorough molecular investigation of the altered biochemical mechanisms seems to be a necessary step for deeply understanding the phenotype associated with each individual variant. Clinically similar phenotypes may indeed result from very different molecular features of the affected GCAP1 variant and, conversely, similar molecular properties may result in heterogenous phenotypes [13]. The dominant nature of the transmission in COD associated with *GUCA1A* poses tremendous challenges to possible therapeutic options, because the compensation of the detrimental effects of the mutant allele, which in other species may arise from other GCAP isoforms, does not seem to occur in humans [26].

In this work, we identified a novel variant of GCAP1 associated with COD that shows an unusual clinical phenotype. The electroretinogram is severely altered under both scotopic and photopic conditions and shows a negative pattern with abnormally attenuated b-wave component. Mutations in *GUCA1A* result in the substitution of two adjacent amino acids in the high affinity binding site EF-hand 3 (EF3), namely, asparagine 104 with lysine and glycine 105 with arginine (N104K-G105R), both positions being highly conserved in vertebrates (Appendix A). Although the first substitution was previously observed in a family with adCOD [19], this is the first report of a double amino acid substitution in *GUCA1A* associated with adCOD. The key role of the replaced amino acids in Ca^2+^ coordination, and therefore in switching between GC-activating and inhibiting states, prompted us to express the human variant in a heterologous system and characterize its structural and functional properties in deep detail, by integrating biochemical and biophysical studies with molecular dynamic simulations. We found that the double mutation in GCAP1 leads to a dramatic loss of Ca^2+^ sensitivity, although in the absence of major alterations of protein secondary and tertiary structure. Subtle changes in the Mg^2+^- and Ca^2+^-bound conformations and their flexibility are sufficient to profoundly alter the activation profiles of both GC1 and GC2. We propose that an alteration of the GC1–GCAP1 transduction unit exists at the synaptic terminal as well as in the photoreceptor outer segment, and such alteration may be related with the abnormal synaptic transmission between photoreceptors and bipolar cells.

## 2. Results

### 2.1. Clinical Phenotype and Disease Progression

The proband was a female patient, aged 54 at the time of our first observation, who reported to have no children and no living parents, thus appearing to be an isolate case. She reported photophobia, low vision, and color vision loss since the age of 20. She underwent three clinical examinations and two electroretinogram (ERG) examinations, each one year apart. Her visual acuity was 1.3 LogMAR (not improving with pinhole or lens correction) in both eyes and remained stable in the clinical examinations. 

The patient underwent color vision testing (Ishihara plates) that showed a complete loss of color discrimination, and fundus imaging (optical coherence tomography (OCT) and fundus autofluorescence (FAF), Figure 1). No significant changes in the fundus picture or in the retinal microanatomy were observed during the three-year follow up (Appendix A).

OCT showed a central lesion involving the macula, characterized by thinning of the outer nuclear layer, which appeared absent in the fovea, interruption of the ellipsoid zone [27], and abnormal retinal lamination both in the outer and inner retina. Fundus autofluorescence showed atrophy of retinal pigment epithelium and outer retina in a large area at the posterior pole, expressed by hypoautofluorescence. The atrophy was quantified in terms of OCT retinal thickness and FAF area (Table 1).

Electroretinography was performed on two occasions according to published techniques [28,29]. ERG testing showed an almost undetectable cone-mediated electroretinogram, recorded at both single flashes and 30 Hz flicker, and a reduced amplitude of the mixed rod-cone ERG with a reduced b/a wave ratio. Figure 2 depicts representative examples of rod, mixed rod-cone, and cone ERGs from the patient and a control subject. The patient’s mixed ERG showed a compromised b/a wave ratio and undetectable cone-driven (photopic) ERGs, which did not allow the evaluation of ON–OFF cone responses. The same ERG results were obtained after one year.

Imaging and ERG results led us to confirm the initial clinical diagnosis of COD with negative ERG [30,31]. 

### 2.2. Identification of a Novel Variant in GUCA1A in Heterozygosis

The patient underwent genetic screening analysis to identify possible variants associated with COD. The mean next-generation sequencing (NGS) coverage of targeted bases was 136 X, with 96% covered to at least 25 X. Genetic testing showed a novel heterozygous variant in the *GUCA1A* gene (NM_000409), the c.312_313delinsGC, p.(Asn104_Gly105delinsLysArg) (from now on, N104K-G105R). The deletion/insertion of two adjacent nucleotides resulted in the in-frame replacement of the wild-type amino acids asparagine and glycine with the amino acids lysine and arginine (Appendix A). The variant has been classified by American College of Medical Genetics and Genomics (ACMG) guidelines [32], using the Varsome software as variant of unknown significance (VUS) according to these scores: PM1, located in a mutational hot spot and/or critical and well-established functional domain (e.g., active site of an enzyme) without benign variation; PM2, absent from controls (or at extremely low frequency if recessive) in Exome Sequencing Project, 1000 Genomes Project, or Exome Aggregation Consortium; PP3, multiple lines of computational evidence support a deleterious effect on the gene or gene product (conservation, evolutionary, splicing impact, etc.).

### 2.3. Guanylate Cyclase Regulation by N104K-G105R GCAP1

To assess the functional effects of the identified amino acid substitutions at the protein level, the GCAP1 variant was heterologously expressed and purified. We thus probed the functionality of N104K-G105R by monitoring the Ca^2+^-dependent regulation of its molecular targets, namely GC1 and GC2, and comparing the activation profiles with those of wild type (WT) GCAP1. N104K-G105R displayed a small but significant reduction in GC1 activation at low Ca^2+^ (Figure 3A) and a significantly higher cGMP production under inhibiting conditions (high Ca^2+^, Figure 3A). Although the Ca^2+^-dependence of GC1 regulation by N104K-G105R was not completely abolished, its capability to inhibit GC1 at physiological levels of Ca^2+^ was deeply compromised (X-fold = 0.47 vs. 37.4 for the WT, Appendix A), thus resulting in a constitutive activation of the enzyme. Similar conclusions could be drawn with respect to GC2 regulation by GCAP1 variants (Figure 3B), in terms of maximal activation, Ca^2+^-dependence, and residual activity at high Ca^2+^ (X-fold = 0.6 vs. 10.8 for the WT, Appendix A), even though GC2 sensitivity to GCAP1 activation was at least 4.5-fold lower than GC1. 

### 2.4. GCAP1 N104K-G105R Variant Presents Reduced Ca^2+^-Affinity

The N104K-G105R mutations affect the highly conserved residues 5 and 6 of Ca^2+^-binding loop of EF3 (Figure 4A; Appendix A); therefore, we assessed the effects of the loss of Ca^2+^-coordinating Asn carbonyl group and the substitution of a small Gly with positive, bulky Arg on the Ca^2+^ affinity. 

The differential electrophoretic migration of Neuronal Calcium Sensor (NCS) proteins upon ion binding, even under denaturing conditions, can be exploited to evaluate their conformational changes [33] and differences in Ca^2+^-affinity. Although the reason behind such change in electrophoretic mobility has not been completely elucidated, in the absence of ions NCS proteins migrate at their theoretical molecular mass (MM), whereas they exhibit a shift upon Ca^2+^-binding proportional to their affinity [34]. This is also the case for WT GCAP1 [26], which shifts from ~23 kDa in the absence to ~17 kDa in the presence of Ca^2+^ (Figure 4B), and for its variant N104K-G105R, which shows a less prominent shift upon Ca^2+^-binding, suggesting a reduced affinity for Ca^2+^. 

We evaluated the differences in Ca^2+^-affinity of the double mutant with respect to the WT by an assay based on the competition of GCAP1 variants with the chromophoric Ca^2+^-chelator 5,5′Br_2_-BAPTA [35], whose absorption decreases upon ion binding. The pattern of Ca^2+^-titrations of N104K-G105R (Figure 4C, black circles) was found to be almost overlapping the theoretical curve of the chelator (Figure 4C, grey line); thus, no estimate of the individual macroscopic binding constants was possible in this case. Such lack of competition with 5,5′Br_2_-BAPTA led to the conclusion that the apparent K_d_ of N104K-G105R would be comparable if not higher than that of the chelator (K_d_^app^ = 2.3 µM), which would render the double mutant unable to correctly regulate the GC in the physiological Ca^2+^-range (200–600 nM).

### 2.5. Quaternary Structure and Aggregation Propensity of N104K-G105R GCAP1

Several lines of evidence in the last years have indicated that WT GCAP1 is a functional dimer under physiological conditions [36,37]; thus, we conducted analytical gel filtration and Dynamic Light Scattering (DLS) measurements to address the effects of the double point mutation on the quaternary structure of GCAP1, as well as on its aggregation propensity. Analytical Size exclusion Chromatography (SEC) chromatograms (Figure 5A) of the double mutant showed a prominent peak which shifted from 14.58 mL (Figure 5A, blue trace) in the presence of Mg^2+^ (corresponding to a molecular mass (MM) of 38.2 kDa, Table 2) to 14.50 mL (Figure 5A, red trace) in the presence of Ca^2+^ (corresponding to a MM of 40.2 kDa, Table 2), thus suggesting that the double mutant is predominantly a dimer upon cation binding. Even though the apparent MM of N104K-G105R was slightly smaller than the WT independently on the cation bound (45.9 kDa with Mg^2+^ and 47.8 kDa with both Mg^2+^ and Ca^2+^, Table 2), the apparent MM displayed a WT-like behavior when switching from the Mg^2+^-bound to the Ca^2+^-bound form, which is an increase in apparent MM of ~2 kDa. Interestingly, Ca^2+^-bound N104K-G105R displayed a second peak at 13.62 mL (corresponding to a 71.5 kDa MM), pointing towards the presence of higher order oligomeric species, which appeared only as a shoulder of the main peak in the presence of Mg^2+^.

We further investigated the oligomeric properties of the GCAP1 variant by DLS, which showed the presence of a single population with low polydispersion (PdI) both in the presence of Mg^2+^ (PdI = 0.278, Table 2) and Ca^2+^ (PdI = 0.214, Table 2; Figure 5B). Such low polydispersion allowed the estimation of the hydrodynamic diameter of the double variant, which was found to increase from 9.43 nm in the presence of Mg^2+^ to 10.8 nm in the presence of Ca^2+^, in line with analytical gel filtration profiles. Notably, the hydrodynamic diameters of both Mg^2+^-bound and Ca^2+^-bound N104K-G105R were found to be significantly higher than their WT counterparts, which attested to around 6.35 and 6.85 nm (Table 2), respectively. In addition, the increase in hydrodynamic diameter of the double mutant upon Ca^2+^-binding resulted to be almost three-fold larger than that measured for the WT (∆d = 1.37 vs. 0.5 nm), suggesting that the variant is considerably less compact than native GCAP1. 

We tested whether the observed higher hydrodynamic diameter exhibited by N104K-G105R could be attributed to the exposure of extended hydrophobic surfaces to the solvent. The emission fluorescence spectra of the hydrophobic probe 8-Anilinonaphthalene-1-sulfonic acid (ANS) suggest that this was indeed the case (Appendix A). The analysis of the ratio between the maximal ANS fluorescence emission in the presence and in the absence of the protein suggested that the hydrophobic surface of N104K-G105R available for the probe was ~2.5-fold greater than that exhibited by the WT (I_max_/I_ANS_ = 8.4 vs. 3.4 in the absence of ions, 4.8 vs. 2.5 in the presence of Mg^2+^, and 5.0 vs. 2.4 in the presence of Mg^2+^ and Ca^2+^, respectively, Table 3). Moreover, at odds with the WT, the Ca^2+^-bound form of N104K-G105R was found to be more hydrophobic than the Mg^2+^-bound form (I_max_/I_ANS_ = 5.0 vs. 4.8, Table 3) [38]. It is worth recalling that the comparison of the results of analytical SEC and DLS measurements for NCS is not straightforward [39]; therefore, the differences in the observables can be considered meaningful, rather than the absolute numerical quantifications.

The time evolution of the mean count rate (MCR) over 15 h (Figure 5C) of N104K-G105R showed that in the presence of Mg^2+^ (Figure 5C, blue trace), the GCAP1 double mutant undergoes significant oscillations in terms of stability of the colloidal suspension, whereas such oscillations were more limited in the presence of Ca^2+^. Taken together, our results suggest that the double variant is not inclined to aggregate over time, at odds with the E111V mutation, also located in the EF3 motif and showing significant aggregation independently of the presence of cations, with a more prominent effect observed for Mg^2+^ [12]. 

### 2.6. Structural and Stability Changes of N104K-G105R GCAP1 upon Cation Binding

Circular Dichroism (CD) spectroscopy is a valuable low-resolution tool to evaluate conformational changes in NCS proteins upon ion binding, as it provides information about the protein tertiary structure (in the near UV, 250–320 nm), secondary structure (in the far UV, 200–250 nm) and susceptibility to thermal denaturation (following the ellipticity at fixed wavelength).

The presence of two non-conservative mutations in N104K-G105R did not result in protein unfolding, as shown by the typical α-helix protein spectra in the far UV (Figure 6A) and by the fine structure of near UV spectra (Figure 6B). A detailed look at the far UV spectra highlighted subtle differences in the spectral shape with respect to the WT, defined by the ratio between the typical minima at 222 and 208 nm (θ_222_/θ_208_) and the relative variation in ellipticity at 208 nm upon ion binding (∆θ/θ). In the absence of ions, the double mutant exhibited a smaller θ_222_/θ_208_ ratio compared to the WT (0.87 vs. 0.90), but a larger increase upon Mg^2+^-binding (to 0.90 and 0.91, respectively). On the contrary, Ca^2+^ exerted a smaller effect on the secondary structure of the double mutant, as θ_222_/θ_208_ increased only to 0.92, compared to 0.95 of the WT.

On the other hand, the analysis of near UV CD spectra, representing a fingerprint of the protein tertiary structure, highlighted minor differences in the fine structure of the spectrum in the absence of ions with respect the WT [12], mainly in the Phe and Tyr bands (250–280 nm, Figure 6B). Upon Mg^2+^ addition, N104K-G105R exhibited a WT-like behavior, with almost overlapping spectra (Figure 6B, black and blue traces), thus implying no conformational rearrangement consequent to Mg^2+^-binding. At odds with Mg^2+^, Ca^2+^ addition exerted a significant effect on the spectrum, with again the Phe and Tyr bands being the most involved in the decrease in ellipticity, as a result of the typical conformational change exhibited by NCS proteins (Figure 6B, red trace). 

To evaluate the thermal stability of N104K-G105R, we monitored the CD signal at 222 nm, the spectral local minimum displaying the largest variation upon ion addition. In the absence of cations, N104K-G105R was 3 °C less stable than the WT (T_m_ = 51.1 vs. 54.1, Table 3), whereas Mg^2+^ stabilized the structure (∆T = 9.7 °C, Table 3), although without affecting either the secondary or tertiary structure. Interestingly, the Mg^2+^-bound double mutant was 2.8 °C more stable than WT and retained a higher percentage of secondary structure at 96 °C (unfolding percentages were 26.3% vs. 30.8%, respectively, Table 3). As all other human GCAP1 variants previously studied [11,12,38,40], the Ca^2+^-loaded double mutant also displayed an increase in thermal stability (T_m_ = 79.4 °C, Table 3). Nevertheless, such stabilizing effect exerted by Ca^2+^-binding on N104K-G105R was much less pronounced than that of the WT, which did not exhibit a clear folded-to-unfolded transition under the experimental conditions and showed a lower unfolding percentage (30.4% vs. 36.6%).

### 2.7. Flexibility of GCAP1 Variants at Atomistic Resolution Monitored by Molecular Dynamics Simulations

The double substitution N104K-G105R does not result in protein unfolding or major structural differences with respect to the WT; therefore, we ran 300 ns all-atom molecular dynamics (MD) simulations to evaluate subtle conformational and stability changes ascribable to the mutations in both Ca^2+^- and Mg^2+^-bound signaling states. The analysis of structural flexibility by means of the Cα-root-mean-square fluctuation (RMSF) descriptor pointed towards an enhanced plasticity of the backbone of N104K-G105R in both the Mg^2+^-bound and Ca^2+^-loaded forms compared to the WT (Figure 7 and Appendix A). In detail, the increase in flexibility of the GC activating form (Mg^2+^-bound, Figure 7, left panels) involved mainly the N-lobe, particularly the N-terminal helix and the loop connecting the EF1 and EF2 motifs (Appendix A). Such enhanced flexibility in the N-lobe suggested an allosteric effect of the two mutations N104K and G105R (the fifth and sixth residues of the Ca^2+^-binding loop of the EF3 motif, respectively), which are located in the C-lobe. Interestingly, the Mg^2+^ ion bound in the EF2 motif of N104K-G105R displayed a slightly lower RMSF compared to its counterpart in the WT (0.918 vs. 0.887 Å, Appendix A), while the Mg^2+^ ion bound in the EF3 motif exhibited a more remarkable decrease in the RMSF (1.099 vs. 0.952 Å, Appendix A), in line with the enhanced thermal stability of N104K-G105R in the presence of Mg^2+^. At odds with the Mg^2+^-bound form, the GC-inhibiting form (Ca^2+^-loaded, Figure 7, right panels) of the double mutant showed an overall higher RMSF in both N- and C-lobes, mainly affecting the C-terminal and, again, the N-terminal helix (Appendix A). Similarly, Ca^2+^-coordination was found to be considerably affected by the presence of the two mutations, because all three Ca^2+^-ions were less tightly bound than in the WT. This behavior was somehow expected for the EF3 motif due to the presence of the two mutations in the Ca^2+^-binding loop (RMSF = 1.305 vs. 1.002 Å), but not for EF4 (RMSF = 1.236 vs. 0.955 Å), which belongs to the same C-lobe, and even more so for EF2 (RMSF = 1.008 vs. 0.834 Å), located in the N-lobe. Overall, such observations point again to the well-known role of Ca^2+^-ions in stabilizing the structure of NCS proteins and are in line with the decreased thermal stability of Ca^2+^-loaded N104K-G105R.

## 3. Discussion

The perception of complex senses, such as vision, requires precise, sustained, and extremely fast neurotransmission from receptors to downstream neurons. The presence of ribbon synapses in photoreceptors distinguishes these cells from other neurons because it provides the above features by encoding graded changes of membrane potential induced by light into the modulation of continuous vesicle exocytosis, which permits the transmission of signals to the inner retina [41]. The presynaptic concentration of Ca^2+^ influences the release of synaptic vesicles, and therefore the communication between photoreceptors and horizontal and bipolar cells. This phenomenon has been associated with the dynamic process of disassembly and reassembly of the synaptic ribbon, which appears to be Ca^2+^-dependent, because lowering intracellular Ca^2+^ indeed reduces the synaptic ribbon size, while increases in Ca^2+^ concentration result in the opposite effect [42,43]. While the molecular mechanisms underlying this process remain largely unknown, it has been proven that GCAP2, a paralog of GCAP1, is involved in the disassembly of synaptic ribbons by directly interacting with NADH-complexed RIBEYE, the main protein component of the synaptic ribbon [41,44,45]. Although GCAP1 was shown not to bind to RIBEYE in the bovine system [44], thus excluding its involvement in the dynamic assembly, caution should be taken when comparing these mechanisms in ortholog systems, because significant differences have recently been observed between human and bovine GCAP2 functions, in spite of the high sequence similarity among the species [46]. Beside its major role in the outer segment, GCAP1 has also been localized in the synaptic layer of photoreceptors [47,48]. Although light and electron microscopy showed a prominent localization of GCAP1 in cone outer segments, the protein was also detected in cone inner segments and synaptic regions of human, monkey and bovine retinas [49], thus suggesting a non-phototransduction related function yet to be clarified. 

The localization of the second guanylate cyclase GC2 within photoreceptors is not clear, but it has been established that GC2 is present in much lower amounts compared to GC1 in both bovine [9] and mouse retinas [8], and that GCAP2 is the specific regulator of RetGC2 [50]. Nonetheless, GCAP1 can also stimulate RetGC2 with lower specificity in bovine [51] and mouse retinas [8], and we have now confirmed that this is also the case for the human reconstituted system. This is particularly interesting, in consideration of the following facts; (i) besides the outer segment, retinal GCs have been found to localize in the synaptic layer of the bovine retina [52], showing prominent immunoreactivity in both outer and inner plexiform layers; (ii) a thorough biochemical study showed that the GCAP1–GC1 transduction system present in rod outer segments also exists in presynaptic terminals [53]; (iii) the photoreceptor—bipolar synaptic region also contains a GC1-transduction system that is stimulated by Ca^2+^ [54]. All these lines of evidence suggest that the GCAP1–GC1 complex could be associated with neurosensory processes other than phototransduction. 

Here, we describe a clinical case of COD with negative ERG having a pathologic double missense mutation in the *GUCA1A* gene, resulting in the double substitution N104K-G105R in the translated protein GCAP1. This is the first report of a *GUCA1A*-related disease showing a negative ERG, indicating an abnormality of the ON-pathway related to the b-wave generation, which was not observed even in the N104K single variant [19]. A negative ERG could result from an abnormal synaptic transmission between photoreceptors and bipolar cells [30,31,55,56,57,58,59] altering the ON-pathway in the post-receptor retinal system [60]. Alternatively, it could be the expression of profound retinal remodeling in the inner retina, as observed in other retinal degeneration with different phenotypes and molecular pathologies [61,62]. The presence of undetectable cone ERG with a rod-mediated ERG characterized by a reduced b-wave and b/a wave ratio is, however, more likely to be related to the specific genetic defect, leading to an impaired transmission between photoreceptors and bipolar cells. GCAP1 is expressed in human rod photoreceptors in the outer and inner segment as well as in the synaptic region; therefore, we conclude that the novel N104K-G105R variant may perturb the synaptic physiology due to functional defects.

The molecular analysis of N104K-G105R GCAP1 revealed that the two mutations, similarly to all other human GCAP1 variants previously studied [11,12,24,38,40], do not cause protein unfolding, disruption of the dimeric assembly, or aggregation (Figure 5), but rather they exert subtle effects on the secondary and tertiary structure (Figure 6). Such structural changes occur in an allosteric fashion, as the presence of the two substitutions in the EF3 Ca^2+^-binding motif alters the flexibility of both the N- and C-lobes (Figure 7). Additionally, the substitutions affect the coordination of Mg^2+^ (in the EF2 and EF3 motifs) and Ca^2+^ ions (in all functional EF-hand motifs), although in a different manner. MD simulations indeed suggest that Mg^2+^ is slightly more tightly bound by N104K-G105R than the WT, whereas Ca^2+^ ions exhibit higher fluctuations (Appendix A), in line with the enhanced and decreased thermal stabilities in the presence of Mg^2+^ and Ca^2+^, respectively (Table 3). The confirmation of a looser Ca^2+^-coordination by N104K-G105R was provided by the reduced electrophoretic mobility and by competition assays, which estimated the apparent Ca^2+^-affinity of the double mutant to be comparable or lower than that of the competing chelator 5,5′Br_2_-BAPTA (2.3 µM). Taken together, our results point toward an impaired Ca^2+^-sensitivity of N104K-G105R as the molecular explanation for the pathological phenotype, similarly to D100G/E [20,38], I107T [13] and E111V [12] variants, all located in EF3. Indeed, the hampered Ca^2+^-dependent conformational switch of N104K-G105R, responsible for the inhibition of the GCs, would render the enzymes constitutively active in the physiological Ca^2+^-concentration range (Figure 3).

Such constitutive activation of GC1 in the outer segment, already observed for other COD-associated variants [10,11,12,13,14,15,16,17,18,19,20,21,22,23,24], may result in the dysregulation of the second messenger homeostasis, ultimately leading to the toxic and death-inducing accumulation of both cGMP and Ca^2+^ [63]. However, the peculiar electrical response observed in this patient carrying a novel variant of GCAP1 suggests the transmission to downstream neurons to be compromised, a phenomenon that creates a virtual link from the GC1–GCAP1 transduction complex in the outer segment to that in the synaptic terminal. To date, no physiological role for GC in the retinal synaptic regions has been established, but the combination of our clinical, electrophysiological, and biochemical findings suggest that such a role might exist and needs to be unveiled by future studies.

## 4. Materials and Methods

### 4.1. Patient Studies, Clinical and Ophthalmological Examinations

The proband was a 54-year-old female patient at the time of our first observation in 2018, who underwent three clinical examinations and two ERG examinations, each one year apart between 2018 and 2020, consisting of best corrected visual acuity, anterior segment evaluation by slit-lamp, direct and indirect ophthalmoscopy, and color vision testing by Ishihara plates. Fundus imaging by fundus autofluorescence (FAF) and spectral domain optical coherence tomography (OCT) were performed. 

ERG recordings were obtained following the ISCEV standards. Following a dark-adaptation period of 30 min, Ganzfeld rod-mediated ERGs were recorded in response to white 50 μs flashes of 0.01 cd·s/m^2^. Responses were averaged over 10 stimulus presentations. The interstimulus interval was 2 s. Dark-adapted, mixed rod/cone ERGs were recorded in response to white 50 μs full-field stimuli with an intensity of 3 cd·s/m^2^. The interstimulus interval was 15 s. Five responses were averaged. Following a 20 min adaptation to Ganzfeld bowl light (30 cd/m^2^), Ganzfeld cone-mediated ERGs were recorded in response to white 50 μs full-field stimuli with an intensity of 3 cd·s/m^2^ presented on a steady white background of 30 cd/m^2^. Responses were averaged over 20 stimulus presentations. The interstimulus interval was 1 s. Cone-mediated ERGs were also recorded in response to 30 Hz flicker flashes of 2 cd·s/m^2^ presented on a steady white background of 30 cd/m^2^. Responses were averaged over 20 stimulus presentations. Signals were amplified (50 K), filtered (0.3–250 Hz), digitized at 2 kHz, and averaged with automatic artifact rejection. The baseline to peak rod b-wave amplitude was measured. The amplitudes of the a- and b-waves were measured. Flicker ERG amplitudes were measured peak-to-peak, and their implicit times from the stimulus onset.

In addition, mesopic Ganzfeld ERGs to 1 Hz, 2 cd·s/m^2^ flashes and cone-single flash ERGs to 2 Hz, 2 cd·s/m^2^ flashes on a 20 cd/m^2^ background were recorded according to a recently published technique that was specific for cone and cone-rod dystrophy patient examination [28].

### 4.2. Genetic Testing

Genetic testing was performed at MAGI’S Laboratories (Rovereto, Italy). The proband’s DNA was extracted from saliva (MagPurix Forensic DNA extraction Kit; Zinexts Life Science, Taipei, Taiwan) and analyzed through next-generation sequencing (NGS) on a MiSeq instrument, using the PE 2× 150 bp protocol (Illumina, San Diego, CA, USA) with a custom panel comprising the following genes: ABCA4 (OMIM *601691), ADAM9 (OMIM *602713), AIPL1 (OMIM *604392), C8orf37 (OMIM *614477), CACNA1F (OMIM *300110), CACNA2D4 (OMIM *608171), CDHR1 (OMIM *609502), CEP78 (OMIM *617110), CFAP410 (OMIM *603191), CNGA3 (OMIM *600053), CRX (OMIM *602225), DRAM2 (OMIM *613360), ELOVL4 (OMIM *605512), GUCA1A (OMIM *600364), GUCY2D (OMIM *600179), IFT81 (OMIM *605489), KCNV2 (OMIM *607604), PDE6C (OMIM *600827), PITPNM3 (OMIM *608921), POC1B (OMIM *614784), PROM1 (OMIM *604365), PRPH2 (OMIM *179605), RAB28 (OMIM *612994), RAX2 (OMIM *610362), RIMS1 (OMIM *606629), RPGR (OMIM *312610), RPGR (OMIM *312610), RPGRIP1 (OMIM *605446), SEMA4A (OMIM *607292), TTLL5 (OMIM *612268), UNC119 (OMIM *604011).

NGS raw sequencing data were analyzed using an in-house bioinformatics pipeline, as described elsewhere [64,65]. 

Filtered variants were checked for their novelty and pathogenicity in databases (accessed on 24 November 2018) such as dbSNP (https://www.ncbi.nlm.nih.gov/snp/), ClinVar (https://www.ncbi.nlm.nih.gov/clinvar/), gnomAD (https://gnomad.broadinstitute.org/), and the Human Gene Mutation Database, professional version 2017.2 (https://www.portal.biobase-international.com/hgmd/pro/). 

All variants were also classified for pathogenicity by the American College of Medical Genetics and Genomics (ACMG) classification [32], with the help of the online software Varsome (https://varsome.com/; last consultation date 16 February 2021) [66].

Genetic testing was performed for diagnostic purposes, to confirm the clinical suspicion of cone dystrophy. The proband was invited to sign an informed consent form after appropriate pre-test genetic counseling.

### 4.3. Cloning, Protein Expression and Purification

The cDNA of wild-type human GCAP1-E6S (Uniprot: P43080) was cloned in a pET-11a plasmid between NdeI and NheI restriction sites (Genscript). The E6S variant was introduced to allow post-translational myristoylation on the N-terminal Gly residue by *Saccharomyces cerevisiae* N-myristoyl transferase (yNMT) [67]. The N104K-G105R variants were introduced by site-directed mutagenesis on a pET-11a-GCAP1-E6S plasmid (Genscript).

GCAP1 variants were heterologously expressed in *Escherichia coli* BL21-DE3 cells previously co-transformed with pBB131-yNMT and purified after a sequence of size exclusion chromatography (SEC, HiPrep 26/60 Sephacryl S-200 HR, GE Healthcare) and anionic exchange chromatography (AEC, HiPrep Q HP 16/10, GE Healthcare) as previously described [12,67], with the only modification consisting of pH = 8 in AEC buffers. Quantification of proteins was achieved by Bradford assay [68], using a GCAP1-specific reference curve based on amino acid hydrolysis analysis (Alphalyze). Protein purity was checked on a 15% SDS-PAGE gel, GCAP1 samples were either exchanged against 20 mM Tris-HCl pH 7.5, 150 mM KCl, 1 mM DTT buffer and frozen with liquid nitrogen, or against decalcified 50 mM NH_4_HCO_3_ and lyophilized. Samples were finally stored at −80 °C until use. 

### 4.4. Guanylate Cyclase Assay

Recombinant human guanylate cyclases 1 and 2 (GC1 and GC2, respectively) were expressed in permanent HEK293 cells after transfection with PEI and 10-day selection with geneticin [69]. After cell lysis, cell membranes containing GCs were resuspended in 50 mM HEPES pH 7.4, 50 mM KCl, 20 mM NaCl, 1 mM DTT buffer and incubated with 5 µM GCAP1 variants for 10 min at 30 °C in the presence of 1 mM Mg^2+^ and either 2 mM EGTA (free Ca^2+^ < 19 nM) or ~30 µM Ca^2+^ [26]. Enzymatic reactions were carried out in 30 mM MOPS/KOH pH 7.2, 60 mM KCl, 4 mM NaCl, 1 mM GTP, 3.5 mM MgCl_2_, 0.3 mM ATP, 0.16 mM Zaprinast buffer and blocked with the addition of 50 mM EDTA and boiling. Samples were then centrifuged for 20 min at ~18,000× *g* at 4 °C; finally, the supernatant was loaded on a C18-RP column (Lichrosphere, Merck) to quantify cGMP production. Control experiments were performed in the absence of GCAP1 to measure basal cGMP synthesis for each of the independent replicas. The production of cGMP at low and high Ca^2+^-concentration in the presence of GCAP1 variants was subjected to two-tailed *t*-tests, where the null hypothesis was represented by equal averages between different conditions. All differences were statistically significant (*p*-values < 0.016).

### 4.5. Gel Shift Assay

Cation-specific electrophoretic mobility of GCAP1 variants under denaturing conditions was evaluated on a 15% acrylamide SDS-PAGE gel. GCAP1 variants were diluted to 15 µM in 20 mM Tris-HCl pH 7.5, 150 mM KCl, 1 mM DTT buffer, incubated at 25 °C for 10 min with either 5 mM EDTA, 5 mM EGTA + 1.1 mM Mg^2+^ or 1 mM Mg^2+^ + 5 mM Ca^2+^, boiled for 5 min and loaded on the gel.

### 4.6. Ca^2+^-Affinity Assay

The Ca^2+^-affinity of N104K-G105R was measured by a competition assay with the chromophoric chelator 5,5′Br_2_-BAPTA as previously elucidated [20,34,35]. The lyophilized double mutant was dissolved in 20 mM Tris-HCl pH 7.5, 150 mM KCl, 1 mM DTT, 1 mM Mg^2+^, <50 nM Ca^2+^ buffer, to a concentration of 16.2 µM (measured by Bradford assay [68]) in the presence of 17 µM 5,5′Br_2_-BAPTA. The absorbance signal at 263 nm was monitored after titration with 1/1000 of the initial volume of a 3 mM Ca^2+^ solution at 25 °C until saturation of the theoretical number of binding sites available ([Ca^2+^] > 3 × [protein] + [chelator]). The three independent titrations were fitted to a theoretical model consisting of 3 binding sites and a chelator (5,5′Br_2_-BAPTA Ca^2+^-affinity = 2.3 µM) using CaLigator [70], yielding macroscopic Ca^2+^-binding constants considered unreliable due to the absence of competition. Data reported in Figure 4 were normalized as follows:Normalized Ca^2+^ = [Ca^2+^]/(3 × [protein] + [chelator]),(1)
Relative absorbance = (A_263_ − A_min_)/(Amax − Amin),(2)
where A_263_, A_min_ and A_max_ are the absorbance at 263 nm, the minimum, and the maximum values, respectively.

### 4.7. Analytical Size Exclusion Chromatography

Analytical size exclusion chromatography was performed using the same buffer (20 mM Tris-HCl pH 7.5, 150 mM KCl, 1 mM DTT buffer) and the same calibration curve as in Vallone et al. [71] on a Superose 12 10/300 GL (GE Healthcare). The elution profiles at 280 nm of 42 µM GCAP1-N104K-G105R (200 µl injection volume) collected in the presence of either 500 μM EGTA + 1 mM Mg^2+^ or 500 μM Ca^2+^ + 1 mM Mg^2+^ allowed the identification of the elution volume (V_e_). Apparent MM was estimated from the calibration curve of log(MM) vs. distribution coefficient (D_c_) as described by Vallone et al. [71], in which D_c_ was defined as follows: D_c_ = (V_e_ − V_v_)/(V_t_ − V_v_),(3)
where V_v_ and V_t_ are the void (8 mL) and the total column volume (25 mL), respectively.

### 4.8. Circular Dichroism Spectroscopy and Thermal Denaturation Profiles

Variations in thermal stability, secondary, and tertiary structures of GCAP1 variants were evaluated using a Jasco J-710 spectropolarimeter supplied with a Peltier-type cell holder. GCAP1 samples were diluted to 42 and 10 µM (for near UV and far UV/thermal denaturation, respectively) in 20 mM Tris-HCl pH 7.5, 150 mM KCl, 1 mM DTT buffer, and protein concentration was verified by a Bradford assay [68]. Near UV CD spectra (250–320 nm) of GCAP1 variants were recorded after serial additions of 500 µM EGTA, 1 mM Mg^2+^ and 1 mM Ca^2+^; whereas far UV CD spectra (200–250) were recorded after serial additions of 300 µM EGTA, 1 mM Mg^2+^ and 600 µM Ca^2+^. Spectra were collected after setting 1 nm data pitch, 1 nm bandwidth, 4 s integration time, 50 nm/min scanning speed and 37 °C temperature. Sample composition for thermal denaturation was the same as for far UV spectra; profiles were collected at 222 nm after setting 0.1 °C data pitch, 20–96 °C temperature range, 1 nm bandwidth, 4 s integration time, 90 °C/min scanning speed. Raw data were fitted to the following function:θ_222_ (T) = [(b_f_ + k_f_ × T) + (b_u_ + k_u_ × T) × e^−∆Gt (T)^]/[1 + e^−∆Gt (T)/RT^](4)
where b_f_ and b_u_ are the baseline of the folded and unfolded states, respectively, T is the temperature, k_f_ and k_u_ are the slopes of the respective states, and ∆G_t_ is the variation in Gibbs free energy associated with the native-to-unfolded transition. ∆G_t_ can be further developed in the following equation:∆G_t_ (T) = [∆H × (1 − T/T_m_)] + ∆C_p_ * [T − T_m_ − T × ln (T/Tm)](5)
where ∆H is the variation in enthalpy, ∆C_p_ is the variation in heat capacity at constant pressure, and T_m_ is the melting temperature, whose estimation is reported in Table 3. Data shown in Figure 6 report spectra subtracted with the respective reference spectrum, near UV spectra were subtracted by the mean ellipticity of the 310–320 region (where no signal should be detected) to eliminate noise due to cuvette orientation.

### 4.9. Dynamic Light Scattering

Hydrodynamic diameter and aggregation propensity were measured on a Zetasizer Nano-S (Malvern Instruments) using disposable polystyrene low volume cuvettes and the instrumental setup previously developed by some of the authors [72]. Protein samples were diluted to 42 µM in 20 mM Tris-HCl pH 7.5, 150 mM KCl, 1 mM DTT buffer, added with either 500 µM EGTA + 1 mM Mg^2+^ or 1 mM Mg^2+^ + 500 µM Ca^2+^, filtered with an Anotop 10 (Whatman, cutoff 20 nm), equilibrated for 3 min at 25 °C, and monitored for 15–17 h (400 measurements, each consisting of 12–14 repetitions). Hydrodynamic diameters and PdI for N104K-G105R variant reported in Table 2 are the average ± s.e.m of 400 measurements, results for WT were taken from Dal Cortivo et al. [38].

### 4.10. 8-Anilinonaphthalene-1-Sulfonic Acid Fluorescence Hydrophobicity Assay

The conformational changes upon ion binding of N104K-G105R were evaluated in terms of variations in hydrophobicity, using an assay based on the differential fluorescence emission exhibited by the hydrophobic probe 8-Anilinonaphthalene-1-sulfonic acid (ANS) upon binding to hydrophobic patches. Fluorescence emission spectra (440–600 nm) of 30 µM ANS in 20 mM Tris-HCl pH 7.5, 150 mM KCl, 1 mM DTT buffer were recorded with a Jasco FP-750 spectrofluorometer in the absence and presence of 2 µM N104K-G105R and 500 µM EGTA and after serial additions of 1 mM Mg^2+^ and 1 mM Ca^2+^. Fluorescence excitation was set to 380 nm, bandwidths to 5 nm, number of accumulations to 3; the spectrum of the buffer added with 0.1% DMSO was considered as reference and subtracted.

### 4.11. Molecular Dynamics Simulations

The three-dimensional structure of human Ca^2+^-loaded myristoylated GCAP1 was obtained by homology modeling using the Ca^2+^-bound myristoylated GCAP1 from *Gallus gallus* [73] as template, using a previously described procedure [20]. The mutations N104K-G105R were introduced on WT human mGCAP1 by in silico mutagenesis using Bioluminate (v. 4.0.139, Maestro package v. 12.5.139, Schrödinger) tool “Mutate Residue” and selecting the best proposed rotamer for both sidechains. Mg^2+^-bound human GCAP1 states were modeled after deletion of a Ca^2+^ ion in EF4 and substitution of the Ca^2+^ ion with a Mg^2+^ ion in EF2 and EF3 as previously detailed [74]. All-atom MD simulations were run on the GROMACS 2016.1 simulation package [75] and CHARMM36m [76] force field, where the parameters for the N-terminal myristoylated Gly were manually added (available upon request). All systems underwent energy minimization and equilibration using the same parameters and protocols as previously described [77], while the production phase consisted of 300 ns, as in Abbas et al. [11,40], from which the trajectories of the WT are taken for comparison. The analysis of the Cα-RMSF profiles of GCAP1 variants was computed by the “gmx rmsf” function within the GROMACS package.

## Figures and Tables

**Figure 1 ijms-22-04030-f001:**
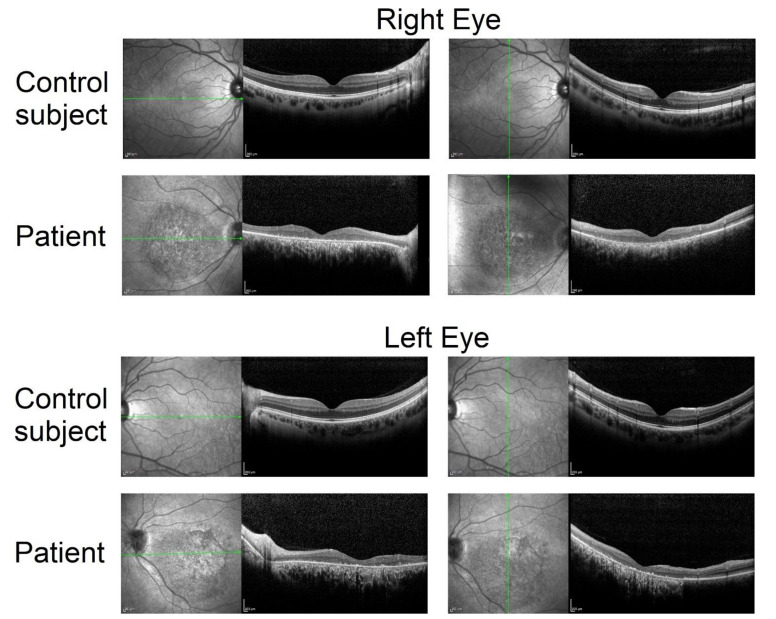
Fundus autofluorescence and optical coherence tomography recorded in the study patient (top rows) and in a healthy control (bottom rows). Green arrows represent the direction of OCT scan. Note the large area of atrophy in the central retina and the thinning of the outer nuclear layer. Scale bars = 200 µm.

**Figure 2 ijms-22-04030-f002:**
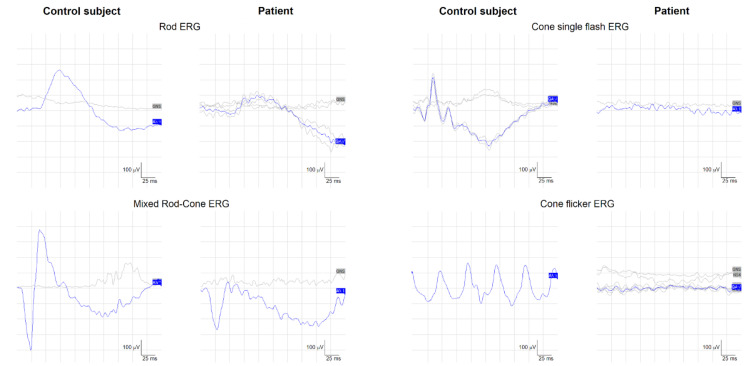
Standard electroretinography recorded in a control subject (a 48-year-old female, left columns) and in the study patient (right columns). Note that cone ERGs were unrecordable in the patient. Rod ERGs showed reduced b-wave amplitude and a lower-than-normal b/a wave amplitude ratio in the dark-adapted mixed rod-cone ERG.

**Figure 3 ijms-22-04030-f003:**
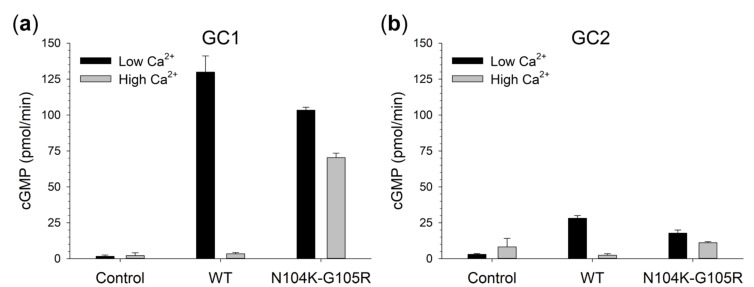
Regulation of guanylate cyclases by GCAP1 variants. Enzymatic regulation of (**a**) human GC1 and (**b**) human GC2 by 5 µM GCAP1 WT and N104K-G105R variants in activating (low Ca^2+^, <19 nM) and inhibiting conditions (high Ca^2+^, ~30 µM). Presented data are an average ± SD of 3 independent experiments.

**Figure 4 ijms-22-04030-f004:**
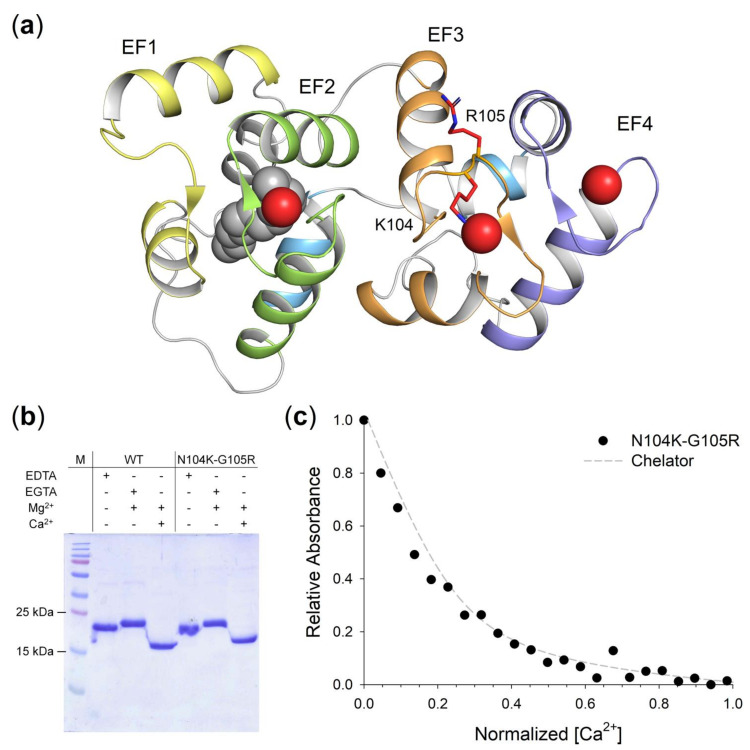
Structural model of N104K-G105R GCAP1 and assessment of Ca^2+^-affinity of GCAP1 variants. (**a**) 3D structural model of human N104K-G105R GCAP1 in the Ca^2+^-loaded form. Protein structure shown in cartoon form, with structural regions specifically colored (N-terminal helix grey, EF1 yellow, EF2 green, EF3 orange, EF4 blue, C-terminal helices cyan). Myristoyl moiety is represented as grey spheres, Ca^2+^ ions as red spheres, residues K104 and R105 as red sticks, with N atoms in blue. (**b**) 15% SDS-PAGE of 20 μM GCAP1 WT and N104K-G105R variants in the presence of 5 mM EDTA, 5 mM EGTA + 1.1 mM Mg^2+^ or 1 mM Mg^2+^ + 5 mM Ca^2+^. (**c**) Example of Ca^2+^ titration curve for N104K-G105R GCAP1. Normalized absorption of 5,5′Br_2_-BAPTA upon Ca^2+^-binding in competition with N104K-G105R GCAP1 (black circles) in the presence of 1 mM Mg^2+^, together with the theoretical simulated curve of the chelator in the absence of competition (grey dashed line). Ca^2+^-concentration considers the dilution upon Ca^2+^-addition, normalization details are reported in the Methods section.

**Figure 5 ijms-22-04030-f005:**
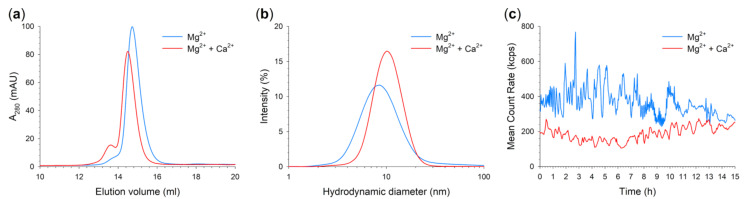
Quaternary structure and aggregation propensity of N104K-G105R monitored by analytical SEC and DLS. (**a**) Analytical SEC chromatograms of 42 µM N104K-G105R in the presence of 500 μM EGTA + 1 mM Mg^2+^ (blue) or 500 μM Ca^2+^ + 1 mM Mg^2+^ (red). (**b**) DLS measurements of 42 µM N104K-G105R in the presence of 500 μM EGTA + 1 mM Mg^2+^ (blue) or 500 μM Ca^2+^ + 1 mM Mg^2+^ (red). (**c**) Time evolution of the mean count rate over 15 h of 42 µM N104K-G105R in the presence of 500 μM EGTA + 1 mM Mg^2+^ (blue) or 500 μM Ca^2+^ + 1 mM Mg^2+^ (red). Analytical SEC and DLS experiments were carried out in 20 mM Tris-HCl pH 7.5, 150 mM KCl, 1 mM DTT buffer. Estimations of the apparent MM and the hydrodynamic diameter are reported in Table 2.

**Figure 6 ijms-22-04030-f006:**
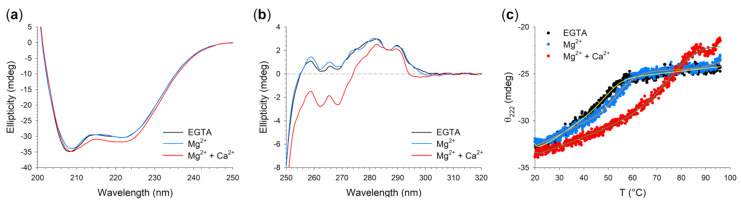
Structural and stability changes of N104K-G105R upon ion binding assessed by Circular Dichroism (CD) spectroscopy. (**a**) Far UV CD spectra of 10 μM N104K-G105R in the presence of 300 μM EGTA (black) and after serial additions of 1 mM Mg^2+^ (blue) and 600 μM Ca^2+^ (red). (**b**) Near UV CD spectra of 42 μM N104K-G105R in the presence of 500 μM EGTA (black) and after serial additions of 1 mM Mg^2+^ (blue) and 1 mM Ca^2+^ (red). (**c**) Thermal denaturation profiles of 10 μM N104K-G105R in the presence of 300 μM EGTA (black), 300 μM EGTA + 1 mM Mg^2+^ (blue) and 1 mM Mg^2+^ + 300 μM free Ca^2+^ (red). CD spectroscopy measurements were conducted in 20 mM Tris-HCl pH 7.5, 150 mM KCl, 1 mM DTT buffer. Thermal denaturation profiles were collected by monitoring the ellipticity at 222 nm between 20 and 96 °C and were fitted to a function accounting for thermodynamic contributions described in the Methods section.

**Figure 7 ijms-22-04030-f007:**
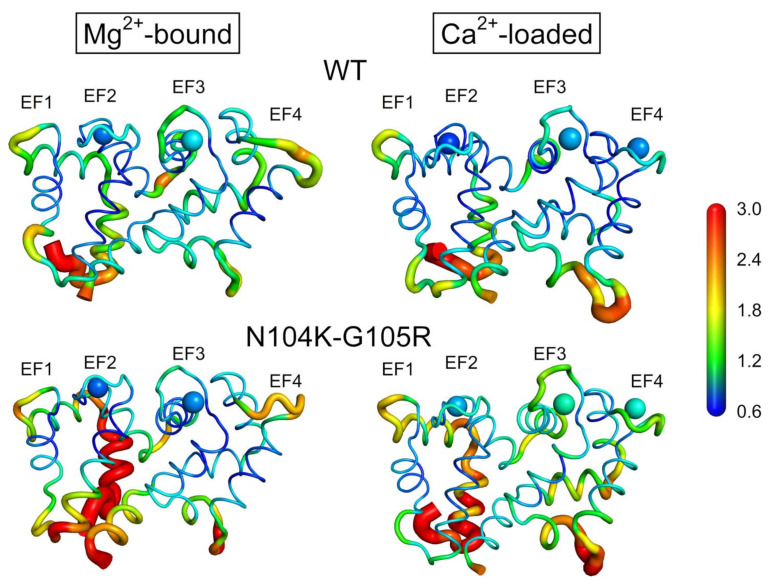
Cα-root-mean-square fluctuation (RMSF) projected on the 3D structure of GCAP1 WT (upper panels) and N104K-G105R (lower panels) in their activating (Mg^2+^-loaded, left) and inhibiting (Ca^2+^-loaded, right) states. Protein structure is displayed as a tube cartoon with diameter proportional to the RMSF; Mg^2+^ and Ca^2+^ ions are shown as spheres. Structures are colored in a rainbow scheme representing RMSF values from 0.6 to 3 Å (see Appendix A for Cα-RMSF profiles).

**Table 1 ijms-22-04030-t001:** Quantification of the evolution of the retinal pigment epithelium atrophy by optical coherence tomography (OCT) and fundus autofluorescence (FAF).

	Optical Coherence Tomography				Fundus Autofluorescence	
	Central Macular Thickness		Ellipsoid Zone Length		Area of Atrophy	
Year	RE ^a^	LE ^b^	RE ^a^	LE ^b^	RE ^a^	LE ^b^
2018	144	61	0	0	5.77 mm^2^	8.35 mm^2^
2019	143	n.a.	0	0	4.72 mm^2^	7.71 mm^2^
2020	97	105	0	0	5.97 mm^2^	8.45 mm^2^

^a^ RE: right eye; ^b^ LE: left eye.

**Table 2 ijms-22-04030-t002:** Variations of the apparent molecular mass (MM) and hydrodynamic diameter of GCAP1 variants assessed by analytical Size Exclusion Chromatography (SEC) and Dynamic Light Scattering (DLS) measurements.

Variant	Condition	MM ^a^ (kDa)	d ^b^ (nm)	*n* ^c^	PdI ^d^
WT ^e^	Mg^2+^	45.9	6.35 ± 0.07	27	0.342 ± 0.02
	Mg^2+^ + Ca^2+^	47.8	6.85 ± 0.17	20	0.306 ± 0.02
N104K-G105R	Mg^2+^	38.2	9.43 ± 0.07	400	0.278 ± 0.002
	Mg^2+^ + Ca^2+^	40.2	10.80 ± 0.05	400	0.214 ± 0.002

^a^ MM was estimated according to the procedure elucidated in the Methods section; ^b^ d (nm) is the average hydrodynamic diameter ± s.e.m.; ^c^
*n* is the number of measurements; ^d^ PdI is the average polydispersion index ± s.e.m. ^e^ data were taken from Dal Cortivo et al. [38].

**Table 3 ijms-22-04030-t003:** Effects of cation binding on secondary structure, hydrophobicity, and thermal stability of GCAP1 variants monitored by Circular Dichroism (CD) spectroscopy.

Variant	Condition	I_max_/I_ANS_ ^a^	θ_222_/θ_208_ ^b^	Δθ/θ (%) ^c^	T_m_ (°C) ^d^	Unfolding (%) ^e^
WT	EGTA	3.4 ^f^	0.90 ^f^	-	54.1 ^g^	24.6 ^g^
	Mg^2+^	2.5 ^f^	0.91 ^f^	2.8 ^f^	58.0 ^g^	30.8 ^g^
	Mg^2+^ + Ca^2+^	2.4 ^f^	0.95 ^f^	7.7 ^f^	>96 ^g^	30.4 ^g^
N104K-G105R	EGTA	8.4	0.87	-	51.1	24.6
	Mg^2+^	4.8	0.90	0.1	60.8	26.3
	Mg^2+^ + Ca^2+^	5.0	0.92	4.6	79.4	36.6

^a^ I_max_/I_ANS_ is the ratio between the maximal intensity of ANS fluorescence emission in the presence and in the absence of the protein, respectively; ^b^ θ_222_/θ_208_ is the ratio between the ellipticity at 222 and 208 nm; ^c^ Δθ/θ is calculated as (θ_222_^ion^ − θ_222_^EGTA^)/θ_222_^EGTA^; ^d^ T_m_ is the melting temperature estimated after fitting ellipticity at 222 nm to the function detailed in the Methods section; ^e^ unfolding percentage is calculated as (θ_222_^96^ − θ_222_^20^)/θ_222_^20^. ^f^ raw data are taken from Dal Cortivo et al. [38]; ^g^ data are taken from Marino et al. [12].

## Data Availability

The data reported in this work are available upon request from the corresponding authors and are not available to the public because of their size.

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
