# Peer review of "Impaired Ca2+ Sensitivity of a Novel GCAP1 Variant Causes Cone Dystrophy and Leads to Abnormal Synaptic Transmission Between Photoreceptors and Bipolar Cells"

_ijms, 2021, doi:10.3390/ijms22084030_

Round 1

Reviewer 1 Report

The manuscript entitled «Impaired Ca2+ 2 sensitivity of a novel GCAP1 variant causes cone dystrophy and leads to abnormal synaptic transmission between photoreceptors and bipolar cells» by Marino and colleagues describes both clinical and experimental data of the effects of a double amino acid substitution in GCAP1. The mutation leads to macular degeneration with a defective ERG response for both rods and cones, linked to severe calcium binding alterations in the modified protein. My comments are attached below.

The figures are compelling, but they lack both comparative data and scales for non-specialized readers. In the OCT data (figure 1), it is hard to actually see what the authors claim, even when summarized in tabular form. Could they highlight the macular dropout, maybe by adding control retina. And the ERGs in figure 2, I see no technical details (flash and flicker light intensity, scotopic/photopic conditions?) or scale bars. According to the methods section these are Ganzfeld ERGs, any possibility of obtaining multifocal ERG data to pinpoint macular vs peripheral cone function loss? Given the only low expression of GCAP1 in rod outer segments, how do the authors interpret the clear rod deficit? Could this be an « indirect » effect due to late stage degeneration involving bystander rod loss?

The detailed molecular analyses of mutant GCAP1 structure and activity are very nicely performed and illustrated.

My only other comment concerns the hypothesis about how GCAP1 may also have a role in synaptic transmission. Although the authors advance this as surprising, to this reviewer it is not unexpected, these are all soluble proteins and other proteins linked to phototransduction (eg. cone arrestin, although admittedly this may not be strictly speaking phototransduction-related) are found within the synaptic subcellular compartment. And their discussion section points to pre-existing data on synaptic localization. They state that this is the first GCAP1-related mutation to be associated with a negative ERG, which is almost more surprising. So is this also the first identified mutation in this region, indicating a highly specific phenotype linked to mis-regulated calcium?

In conclusion, the paper is an impressive collection of clinical observations and detailed biochemical experiments, but requires a little more clarification.

Author Response

The manuscript entitled «Impaired Ca2+ 2 sensitivity of a novel GCAP1 variant causes cone dystrophy and leads to abnormal synaptic transmission between photoreceptors and bipolar cells» by Marino and colleagues describes both clinical and experimental data of the effects of a double amino acid substitution in GCAP1. The mutation leads to macular degeneration with a defective ERG response for both rods and cones, linked to severe calcium binding alterations in the modified protein. My comments are attached below.

We are glad that this Reviewer found merit in our work. We have addressed all the points raised as follows.

The figures are compelling, but they lack both comparative data and scales for non-specialized readers. In the OCT data (figure 1), it is hard to actually see what the authors claim, even when summarized in tabular form. Could they highlight the macular dropout, maybe by adding control retina.

Comparative data and scales have been added as follows:

An image of a normal retina showing both autofluorescence and OCT has been added, for comparison, to the image showing the patient’s macular dropout, in the new figure 1.

The old figure 1 is now supplementary figure S1, which was referenced in the main text as follows: “No significant changes in the fundus picture or in the retinal microanatomy were observed during the three-year follow up (Figure S1).” (lines 104-106)

And the ERGs in figure 2, I see no technical details (flash and flicker light intensity, scotopic/photopic conditions?) or scale bars.

Calibration scale bars for the Ganzfeld ERGs amplitudes and implicit times have been added in Figure 2. Technical details on the Ganzfeld flash and flicker ERGs have been added in the methods section as follows:

”ERG recordings were obtained following the ISCEV standards. Following a dark-adaptation period of 30 minutes, Ganzfeld rod-mediated ERGs were recorded in response to white 50-μs flashes of 0.01 cd*s/m2. Responses were averaged over 10 stimulus presentations. Interstimulus interval was 2 seconds. Dark-adapted, mixed rod/cone ERGs were recorded in response to white 50-μs full-field stimuli with an intensity of 3 cd*s/m2. Interstimulus interval was 15 seconds. Five responses were averaged. Following a 20-minute adaptation to Ganzfeld bowl light (30 cd/m2), Ganzfeld cone-mediated ERGs were recorded in response to white 50-μs full-field stimuli with an intensity of 3 cd*s/m2 presented on a steady white background of 30 cd/m2. Responses were averaged over 20 stimulus presentations. Interstimulus interval was 1 second. Cone-mediated ERGs were also recorded in response to 30 Hz flicker flashes of 2 cd*s/m2 presented on a steady white background of 30 cd/m2. Responses were averaged over 20 stimulus presentations. Signals were amplified (50 K), filtered (0.3–250 Hz), digitized at 2 kHz, and averaged with automatic artifact rejection. The baseline to peak rod b-wave amplitude was measured. The amplitudes of the a- and b-waves were measured. Flicker ERG amplitudes were measured peak-to-peak, and their implicit times from the stimulus onset.

In addition, mesopic Ganzfeld ERGs to 1 Hz, 2 cd*s/m2 flashes and cone-single flash ERGs to 2 Hz, 2 cd*s/m2 flashes on 20 cd/m2 background were recorded according to a recently published technique that was specific for cone and cone-rod dystrophy patients’ examination [28].” (lines 452-471)

According to the methods section these are Ganzfeld ERGs, any possibility of obtaining multifocal ERG data to pinpoint macular vs peripheral cone function loss?

We attempted to record multifocal ERGs, but the data were unreliable due to the unstable fixation presented by the patient. Since no cone function was detectable by full-field flash cone-driven ERGs (both single flash and flicker), it is likely that both central and peripheral cone function was undetectable in this patient.

Given the only low expression of GCAP1 in rod outer segments, how do the authors interpret the clear rod deficit? Could this be an « indirect » effect due to late stage degeneration involving bystander rod loss?

The Reviewer is raising a very interesting point. Indeed, localization of GCAP1 in primates and human photoreceptors is especially prominent in cone outer segments (see refs. [47,48]). The hypothesis of rod deficit being an indirect effect due to later degeneration is plausible, but since no direct experimental evidence is available, we prefer not to speculate on this aspect.

The detailed molecular analyses of mutant GCAP1 structure and activity are very nicely performed and illustrated.

We are glad that this Reviewer appreciated the thoroughness of our biochemical and biophysical characterization

My only other comment concerns the hypothesis about how GCAP1 may also have a role in synaptic transmission. Although the authors advance this as surprising, to this reviewer it is not unexpected, these are all soluble proteins and other proteins linked to phototransduction (eg. cone arrestin, although admittedly this may not be strictly speaking phototransduction-related) are found within the synaptic subcellular compartment. And their discussion section points to pre-existing data on synaptic localization. They state that this is the first GCAP1-related mutation to be associated with a negative ERG, which is almost more surprising. So is this also the first identified mutation in this region, indicating a highly specific phenotype linked to mis-regulated calcium?

We are not sure about what region the Reviewer is referring to with “this also the first identified mutation in this region”. If our interpretation is correct, that is the structural region where the mutation is located, we would like to point out that EF3 is an hotspot for dystrophy-related variants, as residues D100G/E (ref. 20, 38), I107T (ref. 13) and E111V (ref. 12) all presented significantly decreased Ca2+-affinity. In addition, N104K variant showed increased Ca2+-concentration for half-maximal inhibition, with significantly different ERGs (ref. 19) with respect to the novel double variant N104K-G105R, which is indeed the only mutation associated with a negative ERG.

We clarified this point by adding the following sentences:

“, which was not observed even in N104K single variant [19].” (line 404)

and

“…, similarly to D100G/E [20,38], I107T [19] and E111V [12] variants, all located in EF3.” (line 431)

In conclusion, the paper is an impressive collection of clinical observations and detailed biochemical experiments, but requires a little more clarification.

Reviewer 2 Report

Marino and colleagues investigate the pathological consequences of novel GCAP1 mutations in COP patients, using both biochemical and biophysical approaches. Overall I found this manuscript to be very well written, with results supporting the conclusions drawn. Methods are very well described, and significant efforts were taken to characterize these GCAP1 mutations.

I have some minor comments:

1] Proband Pedigree chart should be shown (Suppl. Infomation). Were patient parents and/or siblings also screened for these mutations?

2] Were these biallelic mutations in this proband? 

3] Redundant statement: line 97-102 and line 445-450. Remove one.

4] Figure 2: Control Subject: what was the age, gender, of this control?.

Author Response

Marino and colleagues investigate the pathological consequences of novel GCAP1 mutations in COP patients, using both biochemical and biophysical approaches. Overall I found this manuscript to be very well written, with results supporting the conclusions drawn. Methods are very well described, and significant efforts were taken to characterize these GCAP1 mutations.

We are glad that this Reviewer found merit in our work. We have addressed all the points raised as follows.

I have some minor comments:

1] Proband Pedigree chart should be shown (Suppl. Infomation). Were patient parents and/or siblings also screened for these mutations?

Unfortunately, no relatives were available for sequencing, we clarified in the revised version that it is an isolate case in lines 97-98 as follows:

“The proband was a female patient, aged 54 at the time of our first observation, who reported to have no children and no living parents, thus appearing to be an isolate case.”

2] Were these biallelic mutations in this proband? 

NGS analysis showed that the two variants are on the same reads, so we ruled out that the mutations are biallelic.

3] Redundant statement: line 97-102 and line 445-450. Remove one.

We are sorry for the redundancy, both statements were rephrased as follows:

Lines 97-102: “The proband was a female patient, aged 54 at the time of our first observation, who reported to have no children and no living parents, thus appearing to be an isolate case. She reported photophobia, low vision and color vision loss since the age of 20. She underwent three clinical examinations and two ERG examinations, each one year apart. Her visual acuity was 1.3 LogMAR (not improving with pinhole or lens correction) in both eyes and remained stable in the clinical examinations.”

Lines 445-449: “The proband was a 54-years old female patient at the time of our first observation in 2018 who underwent three clinical examinations and two ERG examinations, each one year apart between 2018 and 2020, consisting of best corrected visual acuity, anterior segment evaluation by slit-lamp, direct and indirect ophthalmoscopy, color vision testing by Ishihara plates.”

4] Figure 2: Control Subject: what was the age, gender, of this control?.

It was a female healthy subject of 48 years of age. Figure 2 legend was modified as follows: “Figure 2. Standard electroretinography recorded in a control subject (a 48-years old female, left columns) and…” (line 146-147).